# Clinical Pharmacists’ Involvement in Pharmacogenomics Testing and Related Services in China

**DOI:** 10.3390/jpm12081267

**Published:** 2022-07-31

**Authors:** Xiaowen Hu, Tong Jia, Xinyi Zhang, Caiying Wu, Yuqing Zhang, Jing Chen, Xiaodong Guan, Luwen Shi, Christine Y. Lu, Xiaoyan Nie

**Affiliations:** 1Department of Pharmacy Administration and Clinical Pharmacy, School of Pharmaceutical Sciences, Peking University, Beijing 100191, China; 2111210044@stu.pku.edu.cn (X.H.); jjiatong@pku.edu.cn (T.J.); 18782201139@163.com (X.Z.); 2111210052@bjmu.edu.cn (C.W.); zhangyuqing@bjmu.cn (Y.Z.); jingchen@bjmu.edu.cn (J.C.); guanxiaodong@pku.edu.cn (X.G.); shilu@bjmu.edu.cn (L.S.); 2International Research Center for Medicinal Administration, Peking University, Beijing 100191, China; 3Department of Population Medicine, Harvard Medical School and Harvard Pilgrim Health Care Institute, Boston, MA 02215, USA; christine_lu@hphci.harvard.edu

**Keywords:** pharmacogenomics testing, clinical pharmacist, implementation, self-assessment of abilities, China

## Abstract

Background: Pharmacogenomics (PGx) testing is increasingly used in clinical practice to optimize drug therapies. This study aims to understand the involvement of clinical pharmacists in PGx testing at tertiary hospitals in China and their self-assessed capacity to deliver such services. Methods: We developed a questionnaire exploring clinical pharmacists’ involvement and self-assessed level of capacity of performing PGx tests. A random sample was obtained from the Pharmaceutical Affairs Management Professional Committee of the Chinese Hospital Association. Results: A total of 1005 clinical pharmacists completed the survey. Of these, 996 (99.1%) had heard of PGx tests and 588 (59.0%) had been involved in PGx testing and related services. Some clinical pharmacists (28.9%) provided PGx services at the rate of “1–5 cases/year” while 21.9% of clinical pharmacists provided PGx services at the rate of “>30 cases/year”. Clinical pharmacists most frequently provided PGx testing for cardiovascular diseases. “Consult relevant guidelines/literature” (90.1%) was the most frequently used method to familiarize oneself with PGx testing. About 60% of the pharmacists considered themselves to have poor or fair capacity to provide PGx testing and related services. Conclusions: More than half of the pharmacists had been involved in PGx testing and related services. However, pharmacists generally had little confidence in their knowledge level of and capacity to provide PGx-related services.

## 1. Introduction

Pharmacogenomics (PGx) derives from pharmacogenetics and studies variations of DNA and RNA characteristics related to drug response [1]. It can explain individual differences in pharmacokinetics and pharmacodynamics and guide personalized medicine in clinical practice to improve drug efficacy and safety. Clopidogrel is a good example that supports the importance of PGx in improving drug efficacy. As the *CYP2C19* genotype affects the formation of clopidogrel active metabolites, people with a moderate and low metabolism of *CYP2C19* who are treated with clopidogrel experience reduced platelet inhibition and increased risks of major adverse cardiovascular and cerebrovascular events [2]. PGx can also help determine dose adjustments, as seen in estimating the starting dose of warfarin based on the different genotypes of *CYP2C9*2, CYP2C9*3*, and *VKORC1**-1639G>A*. [3] Moreover, PGx testing can help avoid adverse drug reactions, such as severe cutaneous adverse reactions (SCAR) due to allele mutations in *HLA-B*58:01* during allopurinol treatment [4], as well as the Stevens–Johnson syndrome and toxic epidermal necrolysis (SJS/TEN) induced by carbamazepine due to *HLA-B*15:02* mutations [5]. Although PGx has many advantages in personalized therapy and enhancing patient outcomes, it has not yet been used extensively in clinical practice partly due to its cost. However, many studies have shown that PGx-guided treatment can be a cost-effective and even a cost-cutting strategy [6,7,8].

By 2020, several international consortia and organizations, including the Pharmacogenomics Knowledge Base (PharmGKB), the Pharmacogenomics Research Network, Clinical Pharmacogenomics Implementation Consortium, European Pharmacogenomics Implementation Consortium, and the Dutch Pharmacogenomics Working Group, have collectively provided a comprehensive evidence base of PGx (encompassing clinical guidelines, drug labels, potentially clinically actionable gene-drug associations, and genotype-phenotype relationships) to help practitioners understand how PGx test results should be used to optimize drug therapies [9]. So far, PharmGKB has collected 168 clinical guideline annotations, 183 pathways, and 810 drug label annotations [10]. In China, the Division of Pharmacogenomics of the Chinese Pharmacological Society was established in 2011, indicating that the research and use of PGx and personalized medicine in China have entered a new era. Since 2014, China’s Food and Drugs Administration and the Health and Family Planning Commission has launched a range of policies around gene sequencing. On 31 July 2015, the Personalized Medical Testing Technology Expert Committee under China’s Health and Family Planning Commission established the “Interim Guidelines of Detection Techniques for Drug Metabolizing Enzymes and Acting Target Genes” and “ the Interim Guidelines of Detection Techniques of Individualized Antineoplastic Therapies”. The committee also put forward that by 2030, China’s investments in precision medicine will reach CNY 60 billion (USD ~8.96 billion), hoping that precision medicine will bring further changes to PGx testing in China through increased funding [11].

Despite growing evidence supporting the feasibility of PGx testing in clinical practice, the use of PGx testing in China remains limited. PGx testing for many drugs is limited to use in laboratories and has not been implemented into routine clinical practice. International research indicated that while most pharmacists remained optimistic about the potential of PGx to advance clinical care, their experience with PGx testing was limited. According to a study in Australia, five pharmacists (23.8%) claimed that they had never heard of one of the following terms: personalized medicine, PGx, or PGx tests [12]. A survey [13] in the U.S. indicated that approximately 34% of the respondents (*n* = 24) had performed PGx tests in their practices. Another report in the Netherlands [14] showed that only 98 respondents (14.7%) reported ordering or recommending a PGx test in the last six months. In Thailand and Malaysia, only 7.3% and 5.8% of respondents had ordered or recommended PGx testing in the past [15,16]. In China, Guo and colleagues [17] studied respondents’ (including physicians, pharmacists, and researchers) knowledge of, attitudes towards, and barriers to PGx testing. They found that over 50% of pharmacists recognized the importance of PGx testing (to their clinical practice) and most respondents affirmed that PGx should be utilized in clinical practice to enhance patient care. However, there is a lack of research on how personalized care and PGx testing are currently carried out at medical institutions and the role of clinical pharmacists in them in China.

China has a hierarchical, three-tier health care system. Tertiary hospitals provide medical care for patients with acute and complicated diseases, while clinical pharmacy services are also mainly provided in tertiary hospitals [18]. Thus, our study explores current practices of clinical pharmacists at tertiary medical institutions across China. We seek to understand the role of clinical pharmacists in PGx testing and personalized treatment services at these institutions and their self-assessed level of knowledge and capacity related to PGx testing. Our study should provide a reference for implementing PGx testing into routine clinical practice.

## 2. Materials and Methods

### 2.1. Questionnaire Development

We developed a questionnaire encompassing questions about clinical pharmacists’ characteristics, personal experience, and self-assessed level of knowledge and capacity related to PGx testing. According to current clinical practice in China and similar research on PGx testing in other countries [19,20,21,22,23,24,25], we developed a preliminary questionnaire and invited four PGx experts to review initial items, provide feedback, and suggest necessary changes. The survey was then piloted among 20 pharmacists not included in the study sample to establish the reliability of the questionnaire. The final survey contained 38 questions and was estimated to take 10–15 min to complete (Appendix B). In this paper, we report findings from 28 questions of the survey focusing on the implementation of PGx tests among clinical pharmacists and competencies related to PGx testing of clinical pharmacists. The remaining 10 questions related to the knowledge and attitude of clinical pharmacists will be presented in another article. Questions were divided into five sections: (1) respondent characteristics; (2) the current practice of PGx testing and related services at the respondent’s practicing hospital; (3) personal experience with PGx testing; (4) sources of information about PGx testing; and (5) self-assessed competencies related to PGx testing. The self-assessed competencies were measured on an integer scale of ten, and the proportion of PGx testing and related services in pharmacists’ daily workload was measured on a percentile scale. If the respondent selected “Volunteer to participate in the study”, any following questions had to be answered. In this paper, we report findings from the survey focusing on the practice experiences with PGx testing and related services and self-assessed competencies towards PGx testing.

### 2.2. Data collection and Population

The sample was obtained from the Pharmaceutical Affairs Management Professional Committee of the Chinese Hospital Association. A random sample of clinical pharmacists meeting the following criteria in the working group of the clinical pharmacist training platform was invited for the survey: (1) practicing in the capital cities and sub-provincial cities of each province, in the capital cities of the five autonomous regions or the four municipalities directly under the Central Government; and (2) working in tertiary hospitals. The first invitation was sent on October 14, 2021, and the questionnaire was then sent to all clinical pharmacists who indicated interest in participating in the study by an online survey platform named “Wenjuanxing” (www.wjx.cn) (accessed on 26 July 2022) for data collection questionnaire surveys. The study was approved by the Institutional Review Boards at Peking University, Beijing, China (IRB 2021100). Respondents who completed the survey received 10 yuan as compensation for their time.

### 2.3. Data Analysis

The analysis was based on survey data from questionnaires collected by Wenjuanxing. The self-assessed competencies score was categorized into four levels as described in a previous study [15]: excellent (score > 8 points), good (score 7–8 points), fair (score 5–6 points), and poor (score < 5 points). Descriptive statistics and categorical variables were summarized. Ordinal logistic regression was used to analyze the interaction item of pharmacists’ involvement in PGx testing and related services and their self-assessed capacity to perform PGx testing. Factors associated with the involvement in PGx testing and the self-assessed capacity were analyzed using the multivariate logistic regression analysis. The results of the multivariate analysis were reported as odds ratios (ORs) with 95% confidence intervals (CIs) and *p*-values. All statistical tests were two-sided, and *p* < 0.05 was considered statistically significant. All statistical analyses were conducted with STATA version 15.1.

## 3. Results

### 3.1. Respondent Characteristics

A total of 1005 clinical pharmacists completed the survey. Of these, 996 (99.1%) had heard of PGx tests while nine (0.9%) had not. 588 (59.0%) had been involved in PGx testing and related services while 408 (41.0%) had not. The following analysis was based on the respondents who had heard about PGx tests (*n* = 996, 99.1%) and who had been involved in PGx testing and related services (*n* = 588, 59.0%) before the survey. The characteristics of the studied groups are shown in Table 1 and Figure 1.

Of the 588 respondents, 27.0% (*n* = 159) were male and more than half aged less than 35 years (*n* = 353, 60.0%). The majority (*n* = 439, 74.7%) had a higher qualification level of master’s degree or above, and 8.0% (*n* = 47) had studied abroad. Most participants had 5–10 years (*n* = 227, 38.6%) or >10 years (*n* = 247, 42.0%) of clinical experience while the most common professional title was the middle-level pharmacist (*n* = 370, 62.9%). When it comes to the attitude toward new technologies, 479 respondents (81.5%) were neutral and chose “aware of the need to change and very comfortable adopting new technologies and adopt new technologies before the average person but need to see evidence of success before adopting”, while 78 (13.3%) held positive attitudes towards new technologies.

The majority of the respondents worked at university-affiliated hospitals (*n* = 384, 65.3%). Of all places where respondents practiced medical care, 22.1% (*n* = 130), 33.7% (*n* = 198), and 44.2% (*n* = 260) located in cities with low, middle, and high level of economic status, respectively. Over half of the respondents (*n* = 327, 55.6%) worked in five departments, namely the departments of Oncology (*n* = 85, 14.5%), Cardiology (*n* = 84, 14.3%), Respiratory Diseases (*n* = 59, 10.0%), ICU (*n* = 54, 9.2%), and Cerebrovascular Diseases and Neurology (*n* = 45, 7.7%).

### 3.2. Pharmacogenomics Tests and Related Services Provided in Hospitals

The availability of PGx testing and related services in the hospitals of the 588 respondents who had been involved in PGx testing and related services and 408 respondents who had not been involved in PGx testing and related services is shown in Table 2, respectively.

Overall, 94.6% of those who had been involved in PGx testing and 81.9% of those who had not been involved in PGx testing indicated that the hospitals where they worked in either provided PGx tests or sent patient samples to a third-party clinical lab (Figure 2). Interestingly, almost four-fifths of the respondents (81.8%, *n* = 481) who had been involved in PGx testing reported that the hospitals where they worked in provided PGx test, while less than half of the respondents (40.0%, *n* = 163) who had not been involved in PGx testing reported that the hospitals where they practiced at provided PGx tests.

Regarding the reason for providing PGx testing in their working hospitals, most respondents who had been involved in PGx testing identified that the primary reasons for PGx testing were “For clinical diagnosis, treatment, and research purpose” (45.1%, *n* = 265) and “Just for clinical diagnosis and treatment purpose” (35.7%, *n* = 210). Meanwhile, among the hospitals that provided PGx tests, the tests were provided mostly in the Pharmacy Department (either alone or with other departments; 66.9%, *n* = 322). Furthermore, 374 (63.6%) of respondents who had been involved in PGx testing suggested that PGx testing was provided by clinical pharmacists as an integral part of personalized pharmacy services compared to only 17.2% of respondents who had not been involved in PGx testing indicating that PGx testing was provided by a clinical pharmacist.

The multivariate analysis for clinical pharmacists’ experience with PGx testing indicated that the clinical pharmacists with the following variables were more likely to involve in PGx testing and related services: male (OR: 1.14; 95%CI: 1.02–1.28; *p* = 0.02), with higher degrees (master’s degree vs. doctoral degree, OR: 1.38 vs. 1.42; 95%CI: 1.20–1.57 vs. 1.17–1.73; *p* = 0.00 vs. 0.00), with a higher professional title (middle level pharmacist vs. associate chief or chief clinical pharmacist, OR: 1.33 vs. 1.37; 95%CI: 1.06–1.67 vs. 1.04–1.79; *p* = 0.01 vs. 0.02), practicing province with higher GDP rank (middle vs. high, OR: 1.16 vs. 1.20; 95%CI: 1.00–1.34 vs. 1.04–1.38; *p* = 0.04 vs. 0.01), and holds a positive attitude towards new technologies (OR: 0.02; 95%CI:1.04–1.82; *p* = 0.02).

### 3.3. Clinical Pharmacists’ Experience with Pharmacogenomics Tests

The top three main content of work for clinical pharmacists who are involved in PGx tests were “Explained results to the patient” (30.9%), “Advised physicians on drug selection, dosage, and monitoring based on the results of PGx testing” (29.7%), and “Explained the results to the doctor” (29.1%).

On average, PGx testing and related services accounted for 16.5% of respondents’ daily workload, with 74.8% (*n* = 440) of respondents indicating that these services accounted for only 0–20% of the daily workload, while a minority of participants (4.4%) indicated that these services accounted for over 60% of the daily workload. More than half of respondents who had any personal experience with PGx testing and related services (52.4%, *n* = 308) reported having been consulted by patients about PGx testing results, and 231 (39.3%) and 49 (8.3%) said that no patients had consulted them or that they did not remember.

In terms of the frequency of cases in PGx testing and related services, Figure 3 shows a U-shaped trend, with 28.9% of the clinical pharmacists choosing “1–5 cases/year” and 21.9% choosing “>30 cases/year”. However, just 12.1%, 7.1%, and 3.2% chose “6–10 cases/year”, “11–20 cases/year”, and “21–30 cases/year”, respectively. The multivariate analysis indicated that the following variables were independently associated with a higher frequency of involvement in PGx testing and related services: doctoral degree (OR: 1.17; 95%CI: 1.03–1.34; *p* = 0.02) and practicing in a province with a middle GDP rank (OR: 1.12; 95%CI: 1.02–1.23; *p* = 0.02).

Among the top ten department with the largest number of clinical pharmacists who had been involved in PGx testing and related services, more than 40% provided PGx testing and related services less than once a month on average (<10 cases/year), except for the pharmacists of the Department of Cardiology and Obstetrics and Gynecology. However, respondents who claimed to provide PGx testing and related services more than 30 times per year were mostly from the Department of Obstetrics and Gynecology (33.3%, *n* = 9) (Appendix A).

The following section asked which drugs the respondents supplied PGx testing and related services for the most frequently; respondents can select multiple drugs based on their experiences. As for providing individual PGx testing, the top five individual PGx tests in our study (Table 3) were for clopidogrel (*CYP2C19*) (57.7%, *n* = 339), warfarin (*CYP2C9, VKORC1, CYP4F2*) (49.7%, *n* = 292), voriconazole (*CYP2C19*) (24.3%, *n* = 143), carbamazepine and phenytoin (*HLA-B*) (21.3%, *n* = 125), and tacrolimus (*CYP3A5*) (19.2%, *n* = 113). After classifying the diseases treated by these drugs, respondents suggested that PGx testing was mostly provided for cardiovascular diseases (*n* = 433, 73.6%). Clinical pharmacists in the Department of Cardiology (*n* = 81) were mostly involved in PGx tests for cardiovascular diseases, followed by Respiratory (Resp) (*n* = 48), and Oncology (Onco) (*n* = 41) (Appendix A).

As for providing multiple-gene PGx panel testing, the top three PGx panels (Table 3) were for commonly used clinical drug testing kits (26.9%, *n* = 158), cardiovascular disease common drugs test kit (25.5%, *n* = 150), and cancer drug gene testing kit (23.5%, *n* = 138). After classifying the diseases treated by these panels, participants were most involved in those for cardiovascular diseases (*n* = 273, 46.4%) too. Few respondents (5.4%, *n* = 32) had never been involved in delivering multiple-gene PGx panel testing. Clinical pharmacists in the Department of Cardiology (Card) (*n* = 68) were mostly involved in PGx tests for cardiovascular diseases, followed by Cerebrovascular Diseases and Neurology (Cere/Neur) (*n* = 32), and Respiratory (Resp) (*n* = 29) (Appendix A).

### 3.4. Resources of Information about PGx Testing

When respondents were interpreting test results or assisting doctors to make clinical treatment decisions according to test results, the majority would “Consult relevant professional guidelines/literature reports” (90.1%, *n* = 530), followed by “Drug labels” (65.5%, *n* = 385), and “Consult medical professional software or APP” (58.8%, *n* = 346).

### 3.5. Self-Assessment of the Competencies Related to Pharmacogenomics Testing

When it comes to self-assessing PGx testing-related abilities, among the total group (*n* = 996), more than 60% of the pharmacists considered themselves to have poor or fair PGx-related abilities (score 0–6; 10 being highest confidence). Clinical pharmacists were most confident in identifying which PGx tests were available at their practicing healthcare facility (mean 5.12 ± 3.09), while they were the least confident in suggesting which drugs need PGx testing (mean 4.79 ± 2.61). Only a small proportion of the pharmacists believed that they were fully equipped (scored > 8) in terms of knowledge related to PGx testing and related services, whether it was to identify PGx tests at their practicing facility (17.5%, *n* = 174), to interpret PGx testing results to physicians and/or patients (11.1%, *n* = 111), to advise physicians on treatment decisions (drug selection, dosage, and monitoring) according to PGx test results (10.7%, *n* = 107), or to recommend PGx tests to physicians and/or patients (12.3%, *n* = 122). Only 84 pharmacists (8.4%) claimed fully equipped to evaluate which drugs require PGx testing. Ordinal logistic regression analysis indicates strong positive associations between the pharmacist‘s 5-item self-assessed level of competency related to PGx testing and whether they had been involved in PGx testing (Appendix A). Moreover, the analysis of experiences (1–10 cases/year, 10–30 cases/year, etc.) and competency shows that respondents who underwent PGx testing more frequently (>30 cases/year) rated their self-competence higher (ORs > 2.80 for each self-competency assessment item) (Appendix A).

The results of the multivariate analysis of the baseline characteristics of respondents involving these five items were showed in the Figure 4. The multivariate analysis suggested that pharmacists practicing in provinces with a high GDP rank or who hold positive attitude towards new technologies were more likely to score himself/herself higher on every item of competency.

## 4. Discussion

Our study aims to understand the involvement of clinical pharmacists in PGx testing and related services at tertiary hospitals in China and their self-assessed capacity to deliver such services. The questionnaire was designed based on past literature and current PGx testing in China. We found that most tertiary medical institutions in China are currently providing PGx testing services, and more than half of clinical pharmacists had been involved in PGx testing and related services. However, regardless of previous experience with PGx testing, pharmacists had little confidence in their knowledge level of and capacity to provide PGx-related services.

The poll was completed by 1005 clinical pharmacists. Of these, 996 (99.1%) had heard of PGx testing, with 588 (59.0%) having been involved in PGx testing and related services, while 408 (41.0%) had not been. Overall, 94.6% (556/558) of those who had been involved in PGx testing and 81.9% (334/408) of those who had not, indicated that the hospitals where they worked in either provided PGx tests or sent patient samples to a third-party clinical lab. Among those who had been involved in PGx service, 28.9% had been involved in 1–5 cases/year, while 21.9% had been involved in more than >30 cases/year. Cardiovascular diseases (*n* = 433, 73.6%), rheumatic immune diseases (*n* = 172, 29.3%), infectious diseases (*n* = 151, 25.7%), and oncology therapy (*n* = 141, 24.0%) are the most involved categories for PGx testing. For approaches to attain information to interpret the test results or assist doctors to make clinical decisions, the vast majority chose “Consult relevant professional guidelines/literature reports” (90.1%, *n* = 530), followed by “Drug labels” (65.5%, *n* = 385) and “Consult medical professional software or APP” (58.8%, *n* = 346). About 60% (344/588) of the pharmacists considered themselves to have poor or fair capacity (i.e., scored 0–6 on the self-assessment) to provide PGx services.

For the accessibility of PGx testing in hospitals, interestingly, respondents who had been involved in PGx service were significantly more likely to work in hospitals that provided PGx tests (81.9% vs. 40.0%). Chinese clinical pharmacists are more likely to participate in PGx testing when their practicing hospitals provide PGx testing. This is different to the model in the United States, where clinical pharmacists are more likely to actively participate in services such as interpreting the results of the test and adjusting the dose of drugs after sending the PGx test to a third-party testing platform and are less influenced by whether PGx tests are available at the hospitals where they work. Our findings are higher than those of a Thailand-based investigation, where 30.9% of the respondents reported that PGx testing was available at their hospital [15]. This might reflect that PGx testing is considerably more accessible at medical institutions in China. As mentioned above, relevant measures were taken to promote access to and use of PGx in China, including the establishment of Division of Pharmacogenomics in 2011, the launch of gene-sequencing-related policies in 2014, and the formulation of the “Interim Guidelines of Detection Techniques for Drug Metabolizing Enzymes and Acting Target Genes” and “the Interim Guidelines of Detection Techniques of Individualized Antineoplastic Therapies” in 2015. Healthcare professionals thus have favorable conditions to perform relevant PGx tests as needed. However, we can see that in 40% of the cases, although the affiliated hospitals offered PGx testing, the respondents were not involved in it. The reason for this may be the limitation of their knowledge and the fact that PGx testing-related service is handled by non-clinical pharmacist healthcare providers in the institution, such as physicians and lab staff.

Our study participants had a higher level of engagement (59.0% of our respondents had been involved in PGx services) compared to other countries (Australia, US, Thailand, Malaysia, Kuwait, and European) where only 5.8% to 34.3% [12,13,14,15,16,24,26] of respondents had been involved in PGx services. The reason for this could be that our respondents are clinical pharmacists from tertiary medical institutions, whereas the data from other countries could be healthcare professionals, community pharmacists, outpatient pharmacists, or even preceptors in school of pharmacy. Clinical pharmacists are a subgroup of hospital pharmacists who have a higher level of training in hospitals and have more opportunities to provide clinical pharmaceutical care than general pharmacists who work in pharmacies, general hospitals, or even schools of pharmacy. Moreover, compared to the frequency of practicing PGx in Japan (25.7%, ≥1/month), 32.3% of our respondents provided PGx services at a rate of more than 10 cases per year. We can reasonably infer that at Chinese medical institutions, PGx testing and relative services have been gradually integrated into routine clinical practice and thus the practice of clinical pharmacies and pharmacists over the years. The study also found that pharmacists in Department of Obstetrics and Gynecology performed PGx testing most frequently, likely because pregnancy is a special physiological condition that requires more individual selection of drugs and doses in order to exclude drugs that may cause harm to the mother or child.

Clinical pharmacists most frequently provided PGx testing services for cardiovascular diseases, which is consistent with the priority given to cardiovascular diseases by current international precision medicine and pharmacogenomics. Notably, our study found that the clinical pharmacists who specialize in one department (for example, the oncology department) do PGx testing for other conditions (Respiratory, CVD). In China, specialist clinical pharmacists are not only trained regarding medication problems of their particular specialty but also drug-related problems of the most commonly used drugs, so from that perspective, they are also more generalists.

Furthermore, the top five individual PGx tests in our study were similar to the results of an international study [27] but different from the results of a study in Japan [28], which suggested that the most frequently used PGx tests were those for irinotecan (96.8%), tacrolimus (86.3%), warfarin (83.9%), azathioprine (76.3%), clopidogrel (74.9%), and carbamazepine (72.4%). The possible reason for this is that the frequency of performing various PGx tests in different countries differed by the frequency of genetic mutations in the local population and reimbursement policies by health insurance schemes. For example, the most compelling PGx evidence comes from non-Caucasian populations from South-East Asia, *HLA-related* testing is less frequently performed in North America where the Asian population is a minority. Meanwhile, the PGx test for *HLA-B*15: 02* has been included in the Universal Health Coverage (UHC) scheme in Thailand and a higher utilization of PGx testing for *HLA-B*15: 02* was observed in these hospitals. In China, since 15 June 2019, the comprehensive reform of medical consumption linkage and the new policy concerning insurance coverage of pathological tests were implemented in Beijing. These policy changes may incentivize other provinces to incorporate more PGx tests into medical insurance schemes and thus promote the uptake of PGx tests.

Turning now to approaches to obtaining information related to PGx testing, our respondents most frequently turned to relevant professional guidelines/literature reports (90.1%), drug labels (65.5%), and professional medical software or APPs (58.8%) for information concerning PGx tests and the interpretation of results, which is consistent with the situation in other countries [13,14,26,28], in general, local guidelines, academic journals, and package inserts are recognized as the most useful sources for learning about PGx testing. China has made considerable endeavors to develop local guidelines on PGx testing and related services to guide future practice PGx tests and clinical decisions that could benefit from PGx testing. Specifically, in the 2016 “Precision Medicine Research” program in China, the “Pharmacogenomics and The Comprehensive Evaluation System for Accurate Drug Use of Chinese People” project led by Professor Cui Yi-min at Peking University First Hospital contributed to establishing the evidence base related to PGx testing in China [29].

Developing local guidelines and promoting the capacity of health professionals to provide PGx services are crucial for enhancing the level of pharmacists’ knowledge about PGx testing and their willingness to provide PGx testing and related services. Gauging from the 5-item self-assessed level of competency in this study, most clinical pharmacists were not confident in their capacity to provide PGx services while clinical pharmacists who had been involved PGx testing generally scored themselves higher on the assessed competencies. This is consistent with previous findings in a study from the United States that preceptors who had used PGx in their practice self-assessed their level of PGx knowledge significantly higher than non-PGx users’ self-assessed knowledge (respective mean values, 2.7 ± 0.8 vs. 1.6 ± 0.7, *p* < 0.001) [13]. Furthermore, in an interaction item analysis of experience and competency ratings for PGx testing, respondents providing PGx testing services more than 30 times per year reported greater confidence in their abilities, so there is reason to believe that clinical pharmacists in China have confidence in providing PGx testing and related services if they are involved enough with the process.

Pharmacist capacity to provide PGx services also varied across countries and by evaluation proxies. For example, our study showed that only 8.43% respondents felt fully competent to identify which drugs needed PGx tests (score > 8), while this proportion ranged from 42.9% in Saudi Arabia [30], 23.5% in Kuwait [24], and 30.8% in Japan [28] to more than half (58.6%) in Europe [26]. Our study respondents were the least confident in the capacity to provide PGx services (score > 8, 8.4–17.5%) across these studies, which can be attributable to several aspects. Firstly, there are more than 200 guidelines on PGx testing in the world, and they have marked the recommended levels of whether some drugs need to be tested for PGx. In addition, the Clinical Pharmacogenomics Implementation Consortium (CPIC) [31] has developed 26 guidelines, which involve 139 genes-drugs information pairs, including 65 gene–drugs that are PGx high-risk. Meanwhile, the FDA has provided specific information regarding therapeutic management for 81 gene–drug pairs, for which there is sufficient supportive scientific evidence [32]. However, very few guidelines related to PGx testing exist in China. Most of these guidelines focus on testing techniques but lack information on whether to recommend PGx testing for a certain drug. There are only 12 drugs in the Chinese package inserts, with limited information on drug–gene relationship and recommendations on PGx testing. The insufficient information provided for the pharmacogenomics aspect of drugs and patient health in China is one of the reasons why pharmacists are not confident in their capacity to provide PGx testing and related services. Secondly, there is a lack of professional training and courses on PGx testing and related services for pharmacy students and in the continuing education of pharmacists. A study [24] showed that the lack of education or training was the biggest obstacle to implementing PGx testing in clinical practice. Therefore, the addition of PGx information to undergraduate courses in schools of pharmacy will improve the knowledge of PGx among pharmacy students. Regular training and seminars for clinical pharmacists at medical institutions will also facilitate the effective use and service provision of PGx testing by clinical pharmacists and help increase pharmacists’ self-efficacy. Last but not least, we defined pharmacists’ confidence as scoring greater than 8 on the self-assessed competencies, while other studies often defined it as a binary or categorical variable. Hence, the lack of confidence of clinical pharmacists in our study as compared with results from other studies may partly stem from methodological factors. However, if we lower the threshold score 8 to 6, still only 25.5% of pharmacists were deemed confident in their capacity to determine which drugs need PGx testing.

Our study has some limitations. First, our respondents were all from tertiary medical institutions in China, therefore, the basic knowledge and academic background of clinical pharmacists in this study are overall higher than average pharmacists. The clinical pharmacists in this study were more likely to participate in PGx testing and related services than pharmacists practicing at medical institutions of other tiers (i.e., primary care facilities and secondary hospitals) in China. Thus, our results do not reflect the involvement of clinical pharmacists in PGx testing at medical institutions of other tiers or the overall Chinese health system. Second, clinical pharmacists who had heard of or been involved in PGx testing and related services before the survey were more likely to participate in our study; thus, the level of engagement of clinical pharmacists in PGx testing observed in our study may be higher than the actual level. Third, the number of clinical pharmacists included in our sample is relatively small, which is likely related to the current low overall number of clinical pharmacists in China and the need to develop clinical pharmacist talent teams. According to our search of the literature on the distribution of clinical pharmacists by department in China, we were unable to find any relevant data, demonstrating that there is still little research in this field in China. As a result, we are unsure whether the distribution of clinical pharmacists in different departments in this study sample is representative, so we will refine this survey and conduct more in-depth studies in this question in the future. Fourth, our study data stemmed from questionnaires and not from field studies, subjects may provide more favorable answers to fit a more socially accepted view [33,34], leading to over-estimations in our results. Finally, the types of individuals- and multiple-gene PGx panel tests involved in this study questionnaire were not exhaustive, and there may be cases where drugs that are frequently tested for PGx were not listed in the questionnaire.

## 5. Conclusions

This study found that many tertiary medical institutions in China are currently providing PGx testing, and more than half of the clinical pharmacist respondents have engaged in PGx testing and related services. However, many were not confident in their knowledge and capacity to provide PGx tests and PGx-related services. Our study results highlight the need to design targeted multifaceted interventions to facilitate the uptake of PGx testing in China. The next step should strive to enhance the confidence of pharmacists to carry out PGx testing and related services.

## Figures and Tables

**Figure 1 jpm-12-01267-f001:**
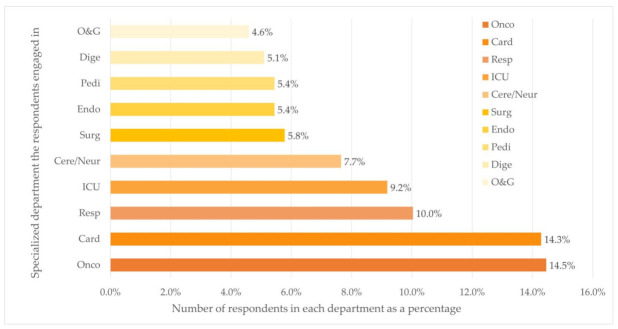
Distribution of department (Top ten) (*n* = 588). Department: Oncology (Onco), Cardiovascular (Card), Respiratory (Resp), Cerebrovascular/Neurology (Cere/Neur), Surgery (Surg), Endocrine (Endo), Pediatrics (Pedi), Digestion (Dige), Obstetrics and Gynecology (O&G).

**Figure 2 jpm-12-01267-f002:**
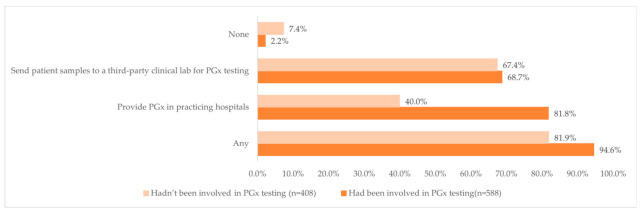
Distribution of the availability of PGx testing at the pharmacists’ work hospital and the clinical pharmacists’ involvement in PGx testing.

**Figure 3 jpm-12-01267-f003:**
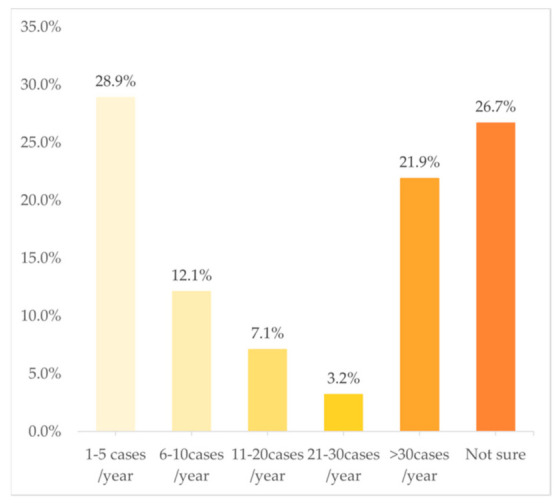
Frequency of work involved in PGx tests (*n* = 588).

**Figure 4 jpm-12-01267-f004:**
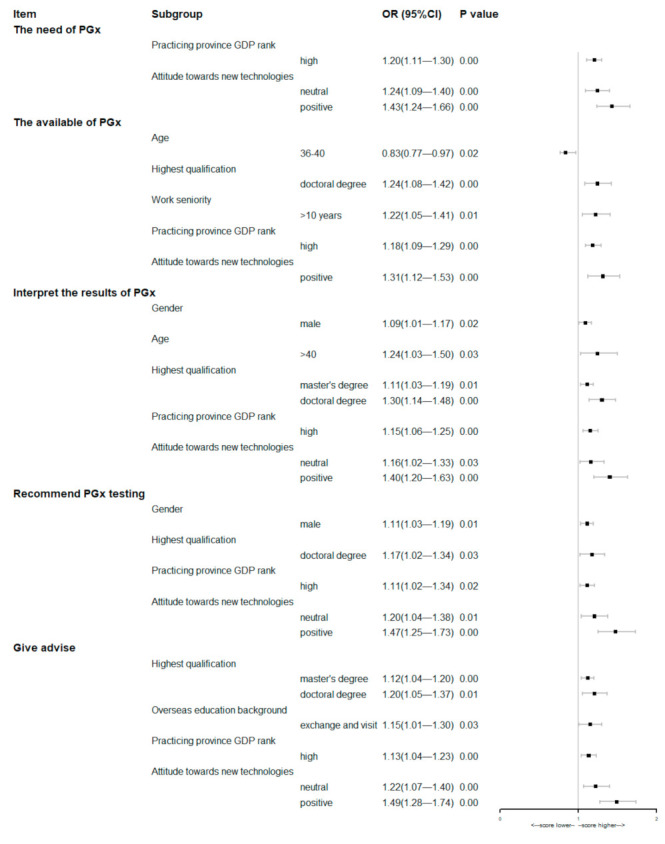
Forest plots of multivariate analyses of respondents’ baseline characteristics associated with the self-assessment of the competencies related to PGx testing (the first option listed in Table 1 was used as the control group, except for gender).

**Table 1 jpm-12-01267-t001:** Characteristics of study respondents.

		Had Heard PGx Tests before	Had Been Involved in PGx Testing and Related Services
		*n* = 996	*n* = 588
Gender			
	male	241 (24.2%)	159 (27.0%)
Age			
	<30	178 (17.9%)	95 (16.2%)
	31–35	468 (47.0%)	258 (43.9%)
	36–40	225 (22.6%)	145 (24.7%)
	>40	125 (12.6%)	90 (15.3%)
Highest qualification			
	bachelor’s degree or below	327 (32.8%)	149 (25.3%)
	master’s degree	592 (59.4%)	381 (64.8%)
	doctoral degree	77 (7.7%)	58 (9.9%)
Overseas education background			
	for a degree	7 (0.7%)	5 (0.9%)
	exchange and visit	50 (5.0%)	42 (7.1%)
Years of clinical experience			
	<5 years	208 (20.9%)	114 (19.4%)
	5–10 years	401 (40.3%)	227 (38.6%)
	10 years	387 (38.9%)	247 (42.0%)
Professional title ^1^			
	junior pharmacist or below	148 (14.5%)	67 (11.4%)
	middle level pharmacist	630 (63.3%)	370 (62.9%)
	associate chief or chief clinical pharmacist	218 (18.3%)	151 (25.7%)
The relationship between practicing hospital and medical college			
	neither affiliated nor teaching hospital	144 (14.2%)	69 (11.7%)
	teaching hospital	213 (21.4%)	124 (21.1%)
	affiliated hospital	609 (61.1%)	384 (65.3%)
GDP rank of the practicing province			
	low	261 (26.2%)	130 (22.1%)
	middle	328 (32.9%)	198 (33.7%)
	high	407 (40.9%)	260 (44.2%)
Attitude towards new technologies			
	negative	71 (7.1%)	31 (5.3%)
	neutral	807 (81.0%)	479 (81.5%)
	positive	118 (11.9%)	78 (13.3%)

Annotation: ^1^ The professional title of hospital pharmacists in China is divided into five levels (from low to high): assistant pharmacist, pharmacist, pharmacist-in-charge, deputy chief pharmacist, and chief pharmacist. In this article, “junior pharmacist or below” includes assistant pharmacist and pharmacist, “middle level pharmacist” is the pharmacist-in-charge, and “associate chief or chief clinical pharmacist” contains the remaining two categories.

**Table 2 jpm-12-01267-t002:** The implementation of pharmacogenomics tests in hospitals (*n* = 996).

Questions of PGx and Related Services Provided in Practicing Hospitals	Had Been Involved in PGx Testing (*n* = 588)	Had Not Been Involved in PGx Testing (*n* = 408)
Whether provide PGx in practicing hospitals?		
Yes	481 (81.8%)	163 (40.0%)
No	94 (16.0%)	215 (52.7%)
Not clear	13 (2.2%)	30 (7.4%)
Purpose of PGx		
For clinical diagnosis, treatment, and research	265 (45.1%)	51 (12.5%)
Just for clinical diagnosis and treatment	210 (35.7%)	103 (25.2%)
Just for research	4 (0.7%)	3 (0.7%)
Not clear	2 (0.3%)	6 (1.5%)
Department that provides PGx tests (Multiple Choice)		
Pharmacy (or with other departments)	322 (66.9%)	63 (15.4%)
Clinical laboratory (or with other departments)	218 (45.3%)	77 (18.9%)
Pathology department (or with other departments)	106 (22.0%)	23 (5.6%)
Others	42 (8.7%)	15 (3.7%)
Not clear	6 (1.2%)	12 (2.9%)
Whether patient samples sent to a third-party clinical lab for PGx testing?		
Yes	404 (68.7%)	275 (67.4%)
No	96 (16.3%)	49 (12.0%)
Not clear	88 (15.0%)	84 (20.6%)
Whether a dedicated person provided personalized pharmacy services based on PGx		
Provided by clinical pharmacists only	374 (63.6%)	70 (17.2%)
Provided by others (physicians or lab workers) only	32 (5.4%)	16 (3.9%)
Provided by clinical pharmacists and others (physicians or lab workers)	35 (6.0%)	6 (1.5%)
Neither	118 (20.1%)	227 (55.6%)
Not clear	29 (4.9%)	89 (21.8%)

**Table 3 jpm-12-01267-t003:** Experience in individual PGx tests and multiple-gene PGx panel tests.

Rank	Type of Work	Freq.	%
Individual PGx Tests			
1	Clopidogrel—*CYP2C19***2* and **3* polymorphism detection	339	57.7
2	Warfarin—*CYP2C9***3* polymorphism; *Vkorc1**-1639 G > A* polymorphism; Detection of *CYP4F2***3* polymorphism	292	49.7
3	Voriconazole—*CYP2C19***2* and *3 polymorphism detection	143	24.3
4	Carbamazepine, phenytoin—*HLA-B* allele detection	125	21.3
5	Tacrolimus—*CYP3A5***3* polymorphism detection	113	19.2
6	Allopurinol—*HLA-B* allele test	95	16.2
7	Antihypertensive drugs—*ACE I/D* polymorphism test; *ADRB1* polymorphism test	94	16.0
8	Trastuzumab—*HER2* gene test	81	13.8
9	Simvastatin, cerivastatin—*SLCO1B1 521T>C* polymorphism detection	76	12.9
10	Irinotecan—*UGT1A1* polymorphism detection	75	12.8
**Multiple-gene PGx panel tests**			
1	Commonly used clinical drug testing kits	158	26.9
2	Cardiovascular disease common drugs test kit	150	25.5
3	Cancer drug gene testing kit	138	23.5
4	Antithrombotic drug test kit	135	23.0
5	Antiepileptic drug test kit	63	10.7
6	Immunosuppressant test kit	60	10.2
7	Anti-infective drug test kit	57	9.7
8	Antihypertensive drug test kit	55	9.4
9	Antihyperlipidemic drugs test kit	54	9.2
10	Commonly used clinical medicines for children	41	7.0
11	Antigout drug test kit	41	7.0

## Data Availability

Not applicable.

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
