# Peer review of "Clinical Pharmacists’ Involvement in Pharmacogenomics Testing and Related Services in China"

_jpm, 2022, doi:10.3390/jpm12081267_

Round 1
Reviewer 1 Report
This study describes the results of questionnaires that were administered to pharmacists in China regarding pharmacogenomic testing. The aim of the study was to understand pharmacists’ experience with and attitudes towards pharmacogenomic testing at tertiary hospitals in China.
In the introduction, the comment about adjusting warfarin starting doses based on “differing mutation rates of CYP2C9*2, CYP2C9*3, and VCORC1-1639G>A mutations in different ethnic groups” is a little odd. Dose adjustments would be due to an individual’s results and not based on different allele frequencies in different ethnic groups. There is also a typo as it should be VKORC1.
It would be helpful to define what is meant by a tertiary hospital in case readers from other counties have different definitions of tertiary hospital.
It would be helpful to define what the different professional titles mean as other countries use different classifications.
Figure 1 needs clear labels, especially the x-axis of the graph.
Results section 3.2, paragraph 2 is difficult to understand. In general, the results section is rather long and includes details that might be better suited for supplemental information, such as table 3. In results section 3.3, the paragraphs describing which disease-areas the respondents supplied PGx testing and related services for the most frequently provides too much information. I would suggest keeping the first bit of information about which disease states were most frequent, but the following information about the individual departments could be removed or moved to a supplement.
Discussion paragraph two simply restates specific results. There’s really no discussion of what those results mean. Lines 399-406 is one run on sentence. The introduction of the top five individual PGx tests in the discussion section might be better in the primary results section. Since part of the aim of the study is to examine PGx testing in China, it would make sense to focus more on the specific type of testing that is being done.
Overall the study provides interesting information about pharmacists’ involvement with PGx testing in China. The methods for data collection and analysis seem appropriate. However, the results section is rather long and includes some repetitive information. I would suggest moving some of the more specifically detailed data to a supplement. The written results are also difficult to follow at times and contain many grammatical errors. Overall this is a good study but the paper requires revisions to focus on the most important data and provide clarity in the results section.
Please see the below list of suggestions of alterations to the current text to improve grammar and typos:
Line 23 “while 21.9% of clinical pharmacists provided PGx services at the rate of “>30 cases/year”
Line 27 “More than half of the pharmacists had been”
Line 38: Gene names should be italicized: ”CYP2C19”
Line 45: “help avoid adverse drug”
Line 47: capitalize “Stevens-Johnson syndrome”
Line 48 “HLA-B*15:02”
Line 107: What is meant by “competencies of them”?
Line 113-114: “any following questions had to be answered”
Line 129: “compensation for their time”
Line 134: “categorical variables were summarized”
Line 149: “(n=588, 59.0%)”
Line 178: “who had not been involved in PGx testing indicated that”
Line 185: “testing identified that the primary reasons for”
Line 188: “in the Pharmacy”
Line 213: “of respondents indicating that these”
Line 252: What is meant by “providing multiple PGx testing”?
Author Response
Dear reviewer:
We appreciate your kind consideration and thorough review of this article. Your comments are very instructive and helpful. we have made comprehensive and detailed revisions according to your suggestions. Please see the attachment.

Reviewer 2 Report
In this manuscript, the authors reported a study on the involvement and capability of Chinese clinical pharmacists in pharmacogenomic (PGx) testing. They developed a questionnaire and distributed it to clinical pharmacists in tertiary hospitals across China. More than one thousand responses were collected. A comprehensive data analysis was performed, and the results are presented in Section 3. Section 4 provides in-depth discussions, including some possible explanations on the results and recommendations on implementing PGx testing.
Overall, this study was well-designed and well-performed. The results of the study were clearly presented and analyzed. The manuscript is well-structured.
Some minor comments related to writing and formatting are as follows:
1. Please make sure that the in-text citation numbers, e.g. “[4]”, and reference numbers, e.g. “Figure S1” are consistently placed, either before punctuations or after punctuations; see Lines 46 and 251 for example.
2. Line 70: please check the conversion between Chinese yuan and USD.
3. Figure 3: Since both number of cases and percentages are marked on each bar, it might be better to introduce a secondary y-axis of percentage.
4. Line 398: please check the language
Author Response
Dear reviewers:
We appreciate your kind consideration and thorough review of this article. Your comments are very instructive and helpful. We have made comprehensive and detailed revisions according to your suggestions. Please see the attachment.

Reviewer 3 Report
Hu et al did a comprehensive study of pharmacogenomics testing in China, and the various features associated with clinical pharmacists recommending this for patients. The article is very well written for the most part. Especially, the introduction and the discussion are top class – great job on this. The synthesis and the development of the questionnaire are commendable. I like how the author distributed and analyzed the gender, age, and educational background of the respondents acquainted with PGx. The data related to overseas education especially as ‘exchange and visit’ impacting’ awareness about PGx was a nice finding. It was also nice to see that a high percentage of hospitals in China are advising patients PGx testing for providing better treatment. I would like the authors to address these comments:
1. Why 28 out of 38 questions were reported? What are the remaining 10 questions? Why were they not considered?
2. The font and sentence spacing are different for some sections: 1 and 2 are the same whereas section 3 is different. Please correct this.
3. Put respectively, on page 4 line 163.
4. Can the departmental association be presented in percentile?
5. It should be a doctoral degree not a doctor’s degree.
6. Table 2 last section “Whether a dedicated person provided personalized pharmacy services based on PGx” what is the difference between the 3rd entry and the first two entries?
7. From figure 2 it appears that, for 40% of cases although affiliated hospitals provide PGx testing, the respondents were not involved in it. Can the authors comment on why is that?
8. Can the authors explain the first paragraph on page 8 so that readers with a little statistical background can also realize it. It would be nice. This also can be applied to the last section on page 8.
9. Page 8 line 207: shouldn’t it be pharmacogenomics?
10. It appears that the Department of Obstetrics and Gynecology although provided the most number of PGx and related services(more than 30 times per year), the same department had the least number of respondents (Figure 1). If a greater number of respondents would be included, how the statistical distribution would get affected?
11. Texts on page 12 had a lot of information and can be overwhelming to read for extracting useful information. Is there a better way to organize these results? For example, like a Circos plot that demonstrates which parameters are weighted more and are much strongly or weakly associated with which parameter? That would be a really nice visual to get the gist of the results without going through all these numbers and statistics in detail.
12. I think the explanation the authors gave about why the level of engagement was higher in China compared to the rest of the world is not very accurate for projection. It just might be the case that, those other studies were based on less number of invitations being sent out or many other reasons. Or as the authors said their respondents were clinical pharmacists, were the respondents clinical pharmacists for these other studies from other countries? I would appreciate the authors’ explanation about this.
13. Can the questionnaire be included in the supplementary document?
Author Response
Dear Reviewer,
We appreciate your kind consideration and thorough review of this article. Your comments are very instructive and helpful. We have, therefore, revised the manuscript accordingly. Please see the attachment.
